# MixFormerV2: Efficient Fully Transformer Tracking

**Yutao Cui**[†]    **Tianhui Song**[†]    **Gangshan Wu**    **Limin Wang**[*]
State Key Laboratory for Novel Software Technology, Nanjing University, China

https://github.com/MCG-NJU/MixFormerV2

## Abstract

Transformer-based trackers have achieved high accuracy on standard benchmarks. However, their efficiency remains an obstacle to practical deployment on both GPU and CPU platforms. In this paper, to mitigate this issue, we propose a fully transformer tracking framework based on the successful MixFormer tracker [14], coined as *MixFormerV2*, without any dense convolutional operation or complex score prediction module. We introduce four special prediction tokens and concatenate them with those from target template and search area. Then, we apply a unified transformer backbone on these mixed token sequence. These prediction tokens are able to capture the complex correlation between target template and search area via mixed attentions. Based on them, we can easily predict the tracking box and estimate its confidence score through simple MLP heads. To further improve the efficiency of MixFormerV2, we present a new distillation-based model reduction paradigm, including dense-to-sparse distillation and deep-to-shallow distillation. The former one aims to transfer knowledge from the dense-head based MixViT to our fully transformer tracker, while the latter one is for pruning the backbone layers. We instantiate two MixForemrV2 trackers, where the **MixFormerV2-B** achieves an AUC of 70.6% on LaSOT and AUC of 56.7% on TNL2k with a high GPU speed of 165 FPS, and the **MixFormerV2-S** surpasses FEAR-L by 2.7% AUC on LaSOT with a real-time CPU speed.

## 1 Introduction

Visual object tracking has been a fundamental and long-standing task in computer vision, which aims to locate the object in a video sequence, given its initial bounding box. It has a wide range of practical applications, which often require for low computational latency. So it is important to *design a more efficient tracking architecture while maintaining high accuracy*.

Recently, the transformer-based one-stream trackers [7, 14, 55] attain excellent tracking accuracy than the previous Siamese-based ones [2, 10, 11], due to the unified modeling of feature extraction and target integration within a transformer block, which allows both components to benefit from the transformer development (e.g. ViT [18], self-supervised pre-training [24] or contrastive pre-training [43]). However for these trackers, *inference efficiency*, especially on CPU, is still the main obstacle to practical deployment. Taking the state-of-the-art tracker MixViT [15] as an instance, its pipeline contains i) **transformer backbone** on the token sequence from target template and search area, ii) **dense corner head** on the 2D search region for regression and iii) **extra complex score prediction module** for classification (i.e., estimating the box quality for reliable online samples selection). To achieve a high-efficiency tracker, there are still several issues on the design of MixViT. First, the dense convolutional corner head still exhibits a time-consuming design, as implied in Tab 1. This is because it densely estimates the probability distribution of the box corners through a total of

---

† Equal contribution. ∗ Corresponding author (lmwang@nju.edu.cn).

| Layer | Head | Score | GPU FPS | GFLOPs |
|-------|------|-------|---------|--------|
| 8 | Pyram. Corner | ✓ | 90 | 27.2 |
| 8 | Pyram. Corner | - | 120(↑ 33.3%) | 26.2 |
| 8 | Token-based | - | 166(↑ 84.4%) | 22.5 |

Table 1: Efficiency analysis on MixViT-B with different heads. 'Pyram. Corner' represents for the pyramidal corner head [15].

| Layer | MLP Ratio | Image Size | CPU FPS | GPU FPS |
|-------|-----------|------------|---------|---------|
| 4 | 1 | 288 | 21 | 262 |
| | | 224 | 30 | 280 |
| | 4 | 288 | 12 | 255 |
| | | 224 | 15 | 275 |
| 8 | 1 | 288 | 12 | 180 |
| | | 224 | 16 | 190 |
| | 4 | 288 | 7 | 150 |
| | | 224 | 8 | 190 |
| 12 | 1 | 288 | 8 | 130 |
| | | 224 | 12 | 145 |
| | 4 | 288 | 4 | 100 |
| | | 224 | 6 | 140 |

Table 2: Efficiency analysis on MixViT-B with different backbone settings. The employed prediction head is plain corner head [14] for the analysis.

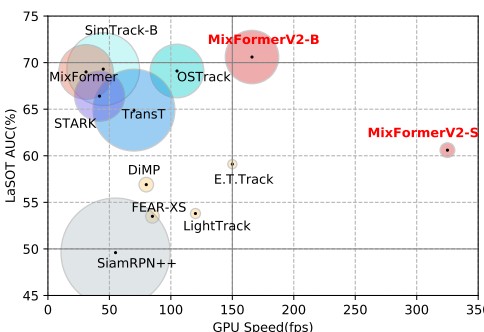

Figure 1: **Comparison with state-of-the-art trackers** in terms of AUC performance, model Flops and GPU Speed on LaSOT. The circle diameter is in proportion to model flops. MixFormerV2-B surpasses existing trackers by a large margin in terms of both accuracy and inference speed. MixFormerV2-S achieves extremely high tracking speed of over 300 FPS while obtaining competitive accuracy compared with other efficient trackers [4, 5].

ten convolutional layers on the high-resolution 2D feature maps. Second, to deal with online template updating, an extra complex score prediction module composed of precise RoI pooling layer, two attention blocks, and a three-layer MLP is required for improving online samples quality, which largely hinders its efficiency and simplicity of MixViT.

To avoid the dense corner head and complicated score prediction module, we propose a new *fully transformer tracking framework—MixFormerV2* without any dense convolutional operation. Our MixFormerV2 yields a very simple and efficient architecture, which is composed of a transformer backbone on the mixed token sequence and two simple MLP heads on the learnable prediction tokens. Specifically, we introduce four special learnable prediction tokens and concate them with the original tokens from target template and search area. Like the CLS token in standard ViT, these prediction tokens are able to capture the complex relation between target template and search area, serving as a compact representation for subsequent regression and classification. Based on them, we can easily predict the target box and confidence score through simple MLP heads, which results in an efficient fully transformer tracker. Our MLP heads directly regress the *probability distribution* of four box coordinates, which improves the regression accuracy without increasing overhead.

To further improve efficiency of MixFormerV2, we present a new model reduction paradigm based on distillation, including *dense-to-sparse distillation* and *deep-to-shallow distillation*. The dense-to-sparse distillation aims to transfer knowledge from the dense-head based MixViT, to our fully transformer tracker. Thanks to the distribution-based regression design in our MLP head, we can easily adopt logits mimicking strategy for distilling MixViT trackers to our MixFormerV2. Based on the observation in Tab. 2, we also exploit the deep-to-shallow distillation to prune our MixFormerV2. We devise a new progressive depth pruning strategy by following a critical principle that constraining the initial distribution of student and teacher trackers to be as similar as possible, which can augment the capacity of transferring knowledge. Specifically, instructed by the frozen teacher model, some certain layers of a copied teacher model are progressively dropped and we use the pruned model as our student initialization. For CPU-realtime tracking, we further introduce an intermediate teacher model to bridge the gap between the large teacher and small student, and prune hidden dim of MLP based on the proposed distillation paradigm.

Based on the proposed model reduction paradigm, we instantiate two types of MixFormerV2 trackers, MixFormerV2-B and MixFormerV2-S. As shown in Fig. 1, MixFormerV2 achieves better trade-off between tracking accuracy and inference speed than previous trackers. Especially, MixFormerV2-B achieves an AUC of 70.6% on LaSOT with a high GPU speed of 165 FPS, and MixFormerV2-S outperforms FEAR-L by 2.7% AUC on LaSOT with a real-time CPU speed. Our contributions are two-fold: 1) We propose the first fully transformer tracking framework without any convolution operation, dubbed as **MixFormerV2**, yielding a more unified and efficient tracker. 2) We present a

new distillation-based model reduction paradigm to make MixFormerV2 more effective and efficient, which can achieve high-performance tracking on platforms with GPUs or CPUs.

## 2 Related Work

**Efficient Visual Object Tracking.** In recent decades, the visual object tracking task has witnessed rapid development due to the emergence of new benchmark datasets[20, 28, 41, 42, 48] and better trackers [2, 10, 12–14, 32, 52, 55]. Researchers have tried to explore efficient and effective tracking architectures for practical applications, such as siamese-based trackers [2, 31, 32, 50], online trackers [3, 16] and transformer-based trackers [10, 37, 52]. Benefiting from transformer structure and attention mechanism, recent works [7, 14, 55] on visual tracking are gradually abandoning traditional three-stage model paradigm, i.e., feature extraction, information interaction and location head. They adopted a more unified one-stream model structure to jointly perform feature extraction and interaction, which turned out to be effective for modeling visual object tracking task. However, some modern tracking architectures are too heavy and computational expensive, making it hard to deploy in practical applications. LightTrack [53] employed NAS to search the a light Siamese network, but its speed was not extremely fast on powerful GPUs. FEAR [5], HCAT [9], E.T.Track [4] designed more efficient framework, however were not suitable for one-stream trackers. We are the first to design efficient one-stream tracker so as to achieve good accuracy and speed trade-off.

**Knowledge Distillation.** Knowledge Distillation [27] was proposed to learn more effective student models with teacher model's supervision. In the beginning, KD is applied in classification problem, where KL divergence is used for measuring the similarity of teacher's and student's predicted probability distribution. For regression problem like object detection, feature mimicking [1, 23, 33] is frequently employed. LD [57] operate logits distillation on bounding box location by converting Dirac delta distribution representation to probability distribution representation of bounding box, which well unifies logits distillation and location distillation. In this work, we exploit some customized strategies to make knowledge distillation more suitable for our tracking framework.

**Vision Transformer Compression.** There exist many general techniques for the purpose of speeding up model inference, including model quantization [22, 47], knowledge distillation [27, 36], pruning [25], and neural architecture search [19]. Recently many works also focus on compressing vision transformer models. For example, Dynamic ViT [44], Evo-ViT[51] tried to prune tokens in attention mechanism. AutoFormer [8], NASViT [21], SlimmingViT [6] employed NAS technique to explore delicate ViT architecture. ViTKD [54] provided several ViT feature distillation guidelines but it focused on compressing the feature dimension instead of model depth. MiniViT [56] applied weights sharing and multiplexing to reduce model parameters. Since one-stream trackers highly rely on training-resource-consuming pre-training, we resort to directly prune the layers of our tracker.

## 3 Method

In this section, we first present the MixFormerV2, which is a more efficient and unified fully transformer tracking framework. Then we describe the proposed distillation-based model reduction, including dense-to-sparse distillation and deep-to-shallow distillation.

### 3.1 Fully Transformer Tracking: MixFormerV2

The proposed MixFormerV2 is a fully transformer tracking framework without any convolutional operation and complex score prediction module. Its backbone is a plain transformer on the mixed token sequence of three types: target template token, search area token, and learnable prediction token. Then, simple MLP heads are placed on top for predicting probability distribution of the box coordinates and corresponding target quality score. Compared with other transformer-based trackers (e.g. TransT [10], STARK [52], MixFormer [14], OSTrack [55] and SimTrack [7]), our MixFormerV2 streamlines the tracking pipeline by effectively removing the customized convolutional classification and regression heads for the first time, which yields a more unified, efficient and flexible tracker. The overall architecture is depicted in Fig. 2. With inputting the template tokens, the search area tokens and learnable prediction tokens, MixFormerV2 predicts the target bounding boxes and quality score in an end-to-end manner.

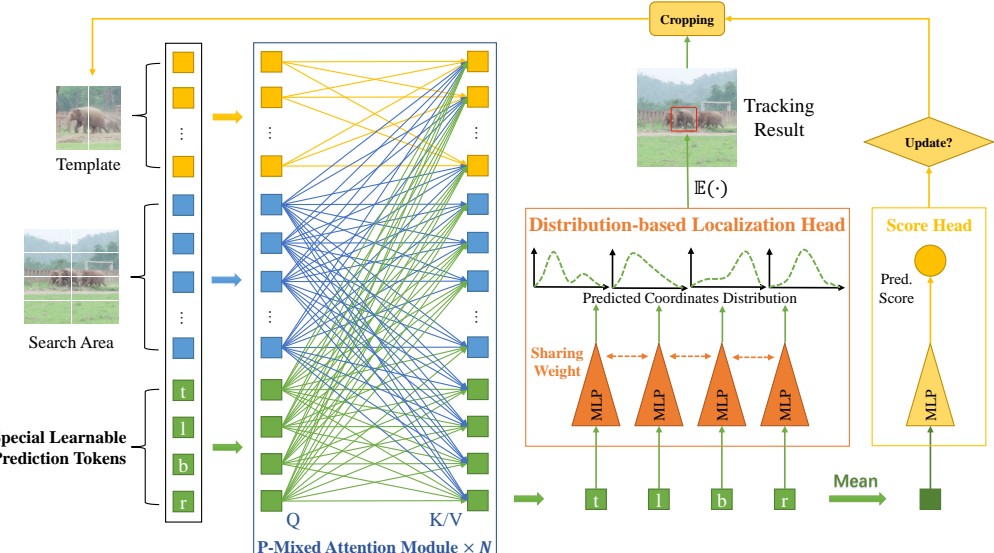

Figure 2: **MixFormerV2 Framework.** MixFormerV2 is a fully transformer tracking framework, composed of a transformer backbone and two simple MLP heads on the learnable prediction tokens.

**Prediction-Token-Involved Mixed Attention.** Compared to original slimming mixed attention [15] in MixViT, the key difference lies in the introduction of the special learnable prediction tokens, which are used to capture the correlation between the target template and search area. These prediction tokens can progressively compress the target information and used as a compact representations for subsequent regression and classification. Specifically, given the concatenated tokens of multiple templates, search and four learnable prediction tokens, we pass them into $N$ layers of prediction-token-involved mixed attention modules (**P-MAM**). We use $q_t$, $k_t$ and $v_t$ to represent template elements (i.e. *query*, *key* and *value*) of attention, $q_s$, $k_s$ and $v_s$ to represent search region, $q_e$, $k_e$ and $v_e$ to represent learnable prediction tokens. The P-MAM can be defined as:

$$k_{tse} = \text{Concat}(k_t, k_s, k_e), \quad v_{tse} = \text{Concat}(v_t, v_s, v_e),$$

$$\text{Atten}_t = \text{Softmax}(\frac{q_t k_t^T}{\sqrt{d}})v_t, \text{Atten}_s = \text{Softmax}(\frac{q_s k_{tse}^T}{\sqrt{d}})v_{tse}, \text{Atten}_e = \text{Softmax}(\frac{q_e k_{tse}^T}{\sqrt{d}})v_{tse}$$

$$(1)$$

where $d$ represents the dimension of each elements, $\text{Atten}_t$, $\text{Atten}_s$ and $\text{Atten}_e$ are the attention output of the template, search and the learnable prediction tokens respectively. Similar to the original MixFormer, we use the asymmetric mixed attention scheme for efficient online inference. Like the CLS tokens in standard ViT, the learnable prediction tokens automatically learn on the tracking dataset to compress both the template and search information.

**Direct Prediction Based on Tokens.** After the transformer backbone, we directly use the prediction tokens to regress the target location and estimate its reliable score. Specifically, we exploit the distribution-based regression based on the four special learnable prediction tokens. In this sense, we regress the probability distribution of the four bounding box coordinates rather than their absolute positions. Experimental results in Section 4.2 also validate the effectiveness of this design. As the prediction tokens can compress target-aware information via the prediction-token-involved mixed attention modules, we can simply predict the four box coordinates with a same MLP head as follows:

$$\hat{P}_X(x) = \text{MLP}(\text{token}_X), X \in \{\mathcal{T}, \mathcal{L}, \mathcal{B}, \mathcal{R}\}. \tag{2}$$

In implementation, we share the MLP weights among four prediction tokens. For predicted target quality assessment, the Score Head is a simple MLP composed of two linear layers. Specifically, firstly we average these four prediction tokens to gather the target information, and then feed the token into the MLP-based Score Head to directly predict the confidence score $s$ which is a real number. Formally, we can represent it as:

$$s = \text{MLP}\left(\text{mean}\left(\text{token}_X\right)\right), X \in \{\mathcal{T}, \mathcal{L}, \mathcal{B}, \mathcal{R}\}.$$

These token-based heads largely reduces the complexity for both the box estimation and quality score estimation, which leads to a more simple and unified tracking architecture.

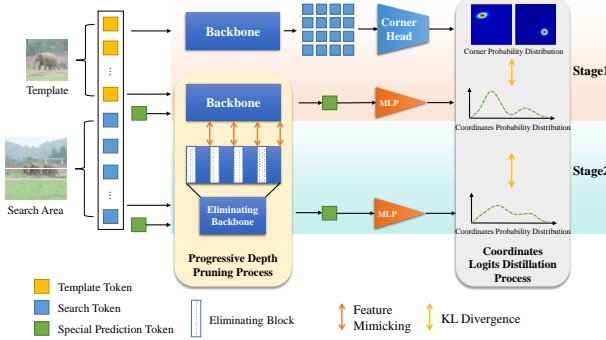

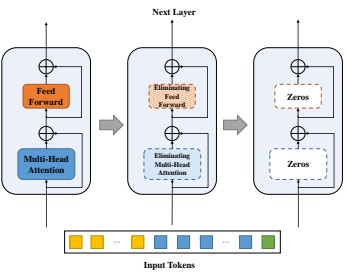

Template

Search Area

Template Token
Search Token
Special Prediction Token

Eliminating Block

Feature Mimicking

KL Divergence

Figure 3: **Distillation-Based Model Reduction** for Mix-FormerV2. The 'Stage1' represents for the dense-to-sparse distillation, while the 'Stage2' is the deep-to-shallow distillation. The blocks with orange arrows are to be supervised and blocks with dotted line are to be eliminated.

Figure 4: **Progressive Depth Pruning Process** for eliminating blocks. All weights in this block decay to zeros and finally only residual connection works, turning into an identity block.

## 3.2 Distillation-Based Model Reduction

To further improve the efficiency and effectiveness of MixFormerV2, we present a distillation-based model reduction paradigm as shown in Fig. 3, which first perform dense-to-sparse distillation for better token-based prediction and then deep-to-shallow distillation for the model pruning.

### 3.2.1 Dense-to-Sparse Distillation

In MixFormerV2, we directly regress the target bounding box based on the prediction tokens to the distribution of four random variables $\mathcal{T}, \mathcal{L}, \mathcal{B}, \mathcal{R} \in \mathbb{R}$, which represents the box's top, left, bottom and right coordinate respectively. In detail, we predict the probability density function of each coordinate: $X \sim \hat{P}_X(x)$, where $X \in \{\mathcal{T}, \mathcal{L}, \mathcal{B}, \mathcal{R}\}$. The final bounding box coordinates $B$ can be derived from the expectation over the regressed probability distribution:

$$B_X = \mathbb{E}_{\hat{P}_X}[X] = \int_{\mathbb{R}} x\hat{P}_X(x)\mathrm{d}x. \tag{3}$$

Since the original MixViT's dense convolutional corner heads predict two-dimensional probability maps, i.e. the joint distribution $P_{\mathcal{TL}}(x, y)$ and $P_{\mathcal{BR}}(x, y)$ for top-left and bottom-right corners, the one-dimensional version of box coordinates distribution can be deduced easily through marginal distribution:

$$
\begin{aligned}
P_{\mathcal{T}}(x) &= \int_{\mathbb{R}} P_{\mathcal{TL}}(x, y)\mathrm{d}y, & P_{\mathcal{L}}(y) &= \int_{\mathbb{R}} P_{\mathcal{TL}}(x, y)\mathrm{d}x \\
P_{\mathcal{B}}(x) &= \int_{\mathbb{R}} P_{\mathcal{BR}}(x, y)\mathrm{d}y, & P_{\mathcal{R}}(y) &= \int_{\mathbb{R}} P_{\mathcal{BR}}(x, y)\mathrm{d}x.
\end{aligned}
\tag{4}
$$

Herein, this modeling approach can bridge the gap between the dense corner prediction and our sparse token-based prediction, i.e., and the regression outputs of original MixViT can be regarded as soft labels for dense-to-sparse distillation. Specifically, we use MixViT's outputs $P_X$ as in Equation 4 for supervising the four coordinates estimation $\hat{P}_X$ of MixFormerV2, applying KL-Divergence loss as follows:

$$L_{\mathrm{loc}} = \sum_{X \in \{\mathcal{T}, \mathcal{L}, \mathcal{B}, \mathcal{R}\}} L_{\mathrm{KL}}(\hat{P}_X, P_X). \tag{5}$$

In this way, the localization knowledge is transferred from the dense corner head of MixViT to the sparse token-based head of MixFormerV2.

### 3.2.2 Deep-to-Shallow Distillation

For further improving efficiency, we focus on pruning the transformer backbone. However, designing a new light-weight backbone is not suitable for fast one-stream tracking. A new backbone of one-stream trackers often highly relies on large-scale pre-training to achieve good performance, which requires for huge amounts of computation. Therefore, we resort to directly cut down some layers

of MixFormerV2 backbone based on both the feature mimicking and logits distillation, as can be seen in Fig. 3:Stage2. Let $F_i^S, F_j^T \in \mathbb{R}^{h \times w \times c}$ denote the feature map from student and teacher, the subscript represents the index of layers. For logits distillation, we use KL-Divergence loss. For feature imitation, we apply $L_2$ loss:

$$L_{\text{feat}} = \sum_{(i,j) \in \mathcal{M}} L_2(F_i^S, F_j^T), \tag{6}$$

where $\mathcal{M}$ is the set of matched layer pairs need to be supervised. Specifically, we design a progressive model depth pruning strategy for distillation.

**Progressive Model Depth Pruning.** Progressive Model Depth Pruning aims to compress MixFormerV2 backbone through reducing the number of transformer layers. Since directly removing some layers could lead to inconsistency and discontinuity, we explore a progressive method for model depth pruning based on the feature and logits distillation. Specifically, instead of letting teacher to supervise a smaller student model from scratch, we make the original student model a complete copy of the teacher model. Then, we will progressively eliminate certain layers of student and make the remaining layers to mimic teacher's representation during training with supervision of teacher. This design allows the initial representation of student and teacher to keep as consistent as possible, providing a smooth transition scheme and reducing the difficulty of feature mimicking.

Formally, let $x_i$ denote output of the $i$-th layer of MixFormerV2 backbone, the calculation of attention block can be represented as below (Layer-Normalization operation is omitted in equation):

$$\begin{aligned}
x_i' &= \text{ATTN}(x_{i-1}) + x_{i-1}, \\
x_i &= \text{FFN}(x_i') + x_i' \\
&= \text{FFN}(\text{ATTN}(x_{i-1}) + x_{i-1}) + \text{ATTN}(x_{i-1}) + x_{i-1},
\end{aligned} \tag{7}$$

Let $\mathcal{E}$ be the set of layers to be eliminated in our student network, we apply a decay rate $\gamma$ on the weights of these layers:

$$x_i = \gamma(\text{FFN}(\text{ATTN}(x_{i-1}) + x_{i-1}) + \text{ATTN}(x_{i-1})) + x_{i-1}, i \in \mathcal{E}. \tag{8}$$

During the first $m$ epochs of student network training, $\gamma$ will gradually decrease from 1 to 0 in the manner of cosine function:

$$\gamma(t) = \begin{cases} 0.5 \times \left(1 + \cos\left(\dfrac{t}{m}\pi\right)\right), & t \leq m, \\ 0, & t > m. \end{cases} \tag{9}$$

This means these layers in student network are gradually eliminated and finally turn into identity transformation, as depicted in Figure 4. The pruned student model can be obtained by simply removing layers in $\mathcal{E}$ and keeping the remaining blocks.

**Intermediate Teacher.** For distillation of an extremely shallow model (4-layers MixFormerV2), we introduce an intermediate teacher (8-layers MixFormerV2) for bridging the deep teacher (12-layers MixFormerV2) and the shallow one. Typically, the knowledge of teacher may be too complex for a small student model to learn. So we introduce an intermediate role serving as teaching assistant to relieve the difficulty of the extreme knowledge distillation. In this sense, we divide the problem of knowledge distillation between teacher and small student into several distillation sub-problems.

**MLP Reduction.** As shown in Table 2, one key factor affecting the inference latency of tracker on CPU device is the hidden feature dim of MLP in Transformer block. In other words, it becomes the bottleneck that limits the real-time speed on CPU device. In order to leverage this issue, we further prune the hidden dim of MLP based on the proposed distillation paradigm, i.e., feature mimicking and logits distillation. Specifically, let the shape of linear weights in the original model is $w \in \mathbb{R}^{d_1 \times d_2}$, and the corresponding shape in the pruning student model is $w' \in \mathbb{R}^{d_1' \times d_2'}$, in which $d_1' \leq d_1, d_2' \leq d_2$, we will initialize weights for student model as: $w' = w[: d_1', : d_2']$. Then we apply distillation supervision for training, letting the pruned MLP to simulate original heavy MLP.

### 3.3 Training of MixFormerV2

The overall training pipeline is demonstrated in Fig. 3, performing dense-to-sparse distillation and then deep-to-shallow distillation to yield our final efficient MixFormerV2 tracker. Then, we train the MLP based score head for 50 epochs. Particularly, for CPU real-time tracking, we employ the intermediate teacher to generate a shallower model (4-layer MixFormerV2) based on the proposed distillation. Besides, we also use the designed MLP reduction strategy for further pruning the CPU real-time tracker. The total loss of distillation training with student $S$ and teacher $T$ is calculated as:

$$L = \lambda_1 L_1(B^S, B^{gt}) + \lambda_2 L_{ciou}(B^S, B^{gt}) + \lambda_3 L_{dist}(S, T), \qquad (10)$$

where the first two terms are exactly the same as original MixFormer's location loss supervised by ground truth bounding box labels, and the rest term is for aforementioned distillation.

## 4 Experiments

### 4.1 Implemented Details

**Training and Inference.** Our trackers are implemented using Python 3.6 and PyTorch 1.7. The distillation training is conducted on 8 NVidia Quadro RTX 8000 GPUs. The inference process runs on one NVidia Quadro RTX 8000 GPU and Intel(R) Xeon(R) Gold 6230R CPU @ 2.10GHz. The training datasets includes TrackingNet [42], LaSOT [20], GOT-10k [28] and COCO [35] training splits., which are the same as MixFormer [14]. Each distillation training stage takes 500 epochs, where the first $m = 40$ epochs are for progressively eliminating layers. We train the score prediction MLP for additional 50 epochs. The batch size is 256, each GPU holding 32 samples. We use AdamW optimizer with weight decay of $10^{-4}$. The initial learning rate is $10^{-4}$ and will be decreased to $10^{-5}$ after 400 epochs. We use horizontal flip and brightness jittering for data augmentation. We instantiate two types of MixFormerV2, including MixFormerV2-B of 8 P-MAM layers for high-speed tracking on GPU platform and MixFormerV2-S of 4 P-MAM layers with MLP ratio of 1.0 for real-time tracking on CPU platform. Their numbers of parameters are 58.8M and 16.2M respectively. The resolutions of search and template images for MixFormerV2-B are $288 \times 288$ and $128 \times 128$ respectively. While for MixFormerV2-S, the resolutions of search and template images are $224 \times 224$ and $112 \times 112$ for real-time tracking on CPU platform. The inference pipeline is the same as MixFormer [14]. We use the first template together with the current search region as input of MixFormerV2. The dynamic templates are updated when the update interval of 200 is reached by default, where the template with the highest score is selected as an online sample.

**Distillation-Based Reduction.** For dense-to-sparse distillation, we use MixViT-L as teacher for training MixFormerV2-B by default. We also try to use MixViT-B as the teacher in Tab 5. Particularly, we employ a customized MixViT-B of plain corner head and with search input size of $224 \times 224$ as the teacher for MixFormer-S. For deep-to-shallow distillation, we use the progressive model depth pruning strategy to produce the 8-layer MixFormerV2-B from a 12-layer one. For MixFormerV2-S, we additionally employs intermediate teacher and MLP reduction strategies, and the process is '12-layers MixFormerV2 to 8-layers MixFormerV2, then 8-layers MixFormerV2 to 4-layers MixFormerV2, finally 4-layers MLP-ratio-4.0 MixFormerV2 to 4-layers MLP-ratio-1.0 MixFormerV2-S'.

### 4.2 Exploration Studies

To verify the effectiveness of our proposed framework and training paradigm, we analyze different components of MixFormerV2 and perform detailed exploration studies on LaSOT [20] dataset.

#### 4.2.1 Analysis on MixFormerV2 Framework

**Token-based Distribution Regression.** The design of distribution-based regression with special learnable prediction tokens is the core of our MixFormerV2. We conduct experiments on different regression methods in Tab. 3a. All models employ ViT-B as backbone and are deployed without distillation and online score prediction. Although the pyramidal corner head obtains the best performance, the running speed is largely decreased compared with our token-based regression head in MixFormerV2. MixFormerV2 with four prediction tokens achieves good trade-off between performance and inference latency. Besides, compared to the direct box prediction with one token on the

| Type | Layer | AUC | FPS |
|---|---|---|---|
| T1 | 12 | 63.1% | 112 |
| T4 | 12 | 67.5% | 110 |
| Py-Corner. | 12 | 69.0% | 92 |

| Method | Score. | AUC | FPS |
|---|---|---|---|
| Ours | - | 68.9% | 166 |
| Ours | ✓ | 70.6% | 165 |
| MixViT-B | - | 69.0% | 92 |
| MixViT-B | ✓ | 69.6% | 80 |

| Stu. | Tea. | Tea-AUC | AUC |
|---|---|---|---|
| base | - | - | 67.5% |
| base | base | 69.0% | 68.9% |
| base | large | 71.5% | 69.6% |

(a) **Different regression methods**. 'T1' denotes direct box prediction based on one token, 'T4' is the proposed distribution-based prediction with 4 prediction tokens, and 'Py-Corner.' is the pyramidal corner head as in MixViT. Models are *without* distillation and score prediction.

(b) **Quality score prediction**. In MixFormerV2, we use token-based MLP head for sample quality score prediction. While in MixViT, it use an extra SPM for score prediction. We employ MixFormerV2-B of 8 layers, *with* the MixViT-L as the distillation teacher, for this analysis.

(c) **Dense-to-Sparse distillation**. The 'base' student denotes the 12-layers MixFormerV2 framework without score prediction. The 'base' teacher is the MixViT-B and the 'large' teacher is the MixViT-L. 'Tea-AUC' is the AUC of the teacher. Models are *without* score prediction.

| Log-dis. | Feat-mim. | AUC |
|---|---|---|
| - | - | 60.7% |
| ✓ | - | 62.4% |
| ✓ | ✓ | 62.9% |

| Init. method | AUC |
|---|---|
| MAE-fir4 | 62.9% |
| Tea-skip4 | 64.4% |
| PMDP | 64.8% |

| Inter. teacher | AUC |
|---|---|
| - | 64.8% |
| ✓ | 65.5% |

(d) **Feature mimicking & logits distillation**. For distillation analysis, we use the MixViT-B of 12 layers with corner head as the teacher, and MixViT of 4 layers as the student.

(e) **Progressive model depth pruning (PMDP)**. 'MAE-fir4' denotes using first 4 layers of MAE-B for student initialization. 'Tea-skip4' is using 4 skipped layers of the teacher.

(f) **Intermediate teacher**. For the analysis, we use the 12-layers MixViT-B as the teacher, 8-layers MixViT as the intermediate teacher and 4-layers MixViT as the student.

| Epoch $m$ | AUC |
|---|---|
| 30 | 68.3% |
| 40 | 68.5% |
| 50 | 68.5% |

| blocks num. | head | AUC |
|---|---|---|
| 12 | Py-Corner. | 69.0% |
| 12 | T4 | 68.9% |
| 8 | T4 | 68.5% |

| blocks num. | head | MLP-r | AUC |
|---|---|---|---|
| 12 | Cor. | 4 | 68.2% |
| 12 | T4 | 4 | 67.7% |
| 8 | T4 | 4 | 66.6% |
| 4 | T4 | 4 | 61.0% |
| 4 | T4 | 1 | 59.4% |

(g) **Eliminating Epochs**. 'Epoch $m$' indicates the number of epochs in progressive eliminating process. The model architecture is based on MixFormerV2-B. Models are *without* score prediction.

(h) **Model pruning route of MixFormerV2-B***. 'T4' denotes the proposed distribution-based prediction with 4 prediction tokens. We use the MixViT-B as the distillation teacher for this analysis.

(i) **Model pruning route of MixFormerV2-S**. 'Cor.' represents for the plain corner head, which is used in the initial teacher model. 'MLP-r' denotes the MLP ratio in attention blocks.

Table 3: **Ablation studies** on LaSOT. The default choice for our model is colored in  gray .

first line of Tab. 3a, which estimates the absolute target position instead of the probability distribution of four coordinates, the proposed distribution-based regression obtains better accuracy. Besides, this design allows to perform dense-to-sparse distillation so as to further boost performance.

**Token-based Quality Score Prediction.** The design of the prediction tokens also allows to perform more efficient quality score prediction via a simple MLP head. As shown in Tab. 3b, the token-based score prediction component improves the baseline MixFormerV2-B by 1.7% with increasing quite little inference latency. Compared to ours, the score prediction module in MixViT-B further decreases the running speed by 13.0%, which is inefficient. Besides, the SPM in MixViT requires precise RoI pooling, which hinders the migration to various platforms.

### 4.2.2 Analysis on Dense-to-Sparse Distillation

We verify the effectiveness of dense-to-sparse distillation in Tab. 3c. When use MixViT-B without its SPM (69.0% AUC) as the teacher model, the MixFormerV2 of 12 P-MAM layers achieves an AUC score of 68.9%, increasing the baseline by 1.4%. This further demonstrate the significance of the design of four special prediction tokens, which allows to perform dense-to-sparse distillation. The setting of using MixViT-L (71.5% AUC) as the teacher model increases the baseline by an AUC score of 2.2%, which implies the good distillation capacity of the large model.

|  | KCF [26] | SiamFC [2] | ATOM [16] | D3Sv2 [38] | DiMP [3] | ToMP [39] | TransT [10] | SBT [49] | SwinTrack [34] | **Ours-S** | **Ours-B** |
|---|---|---|---|---|---|---|---|---|---|---|---|
| **EAO** | 0.239 | 0.255 | 0.386 | 0.356 | 0.430 | 0.511 | 0.512 | 0.522 | 0.524 | 0.431 | **0.556** |
| **Accuracy** | 0.542 | 0.562 | 0.668 | 0.521 | 0.689 | 0.752 | 0.781 | 0.791 | 0.788 | 0.715 | **0.795** |
| **Robustness** | 0.532 | 0.543 | 0.716 | 0.811 | 0.760 | 0.818 | 0.800 | 0.813 | 0.803 | 0.757 | **0.851** |

Table 4: State-of-the-art comparison on VOT2022 [30]. The best results are shown in **bold** font.

### 4.2.3 Analysis on Deep-to-Shallow Distillation

In the following analysis on deep-to-shallow distillation, we use the MixViT-B of 12 layers with plain corner head as the teacher, and MixViT of 4 layers with the same corner head as the student. The models are deployed without score prediction module.

**Feature Mimicking & Logits Distillation.** To give detailed analysis on different distillation methods for tracking, we conduct experiments in Tab. 3d. The models are all initialized with the first 4-layers MAE pre-trained ViT-B weights. It can be seen that logits distillation can increase the baseline by 1.7% AUC, and adding feature mimicking further improves by 0.4% AUC, which indicates the effectiveness of both feature mimicking and logits distillation for tracking.

**Progressive Model Depth Pruning.** We study the effectiveness of the progressive model depth pruning (PMDP) for the student initialization in Tab. 8b. It can be observed that the PMDP improves the traditional initialization method of using MAE pre-trained first 4-layers ViT-B by 1.9%. This demonstrates that it is critical for constraining the initial distribution of student and teacher trackers to be as similar as possible, which can make the feature mimicking easier. Surprisingly, we find that even the initial weights of the four layers are not continuous, i.e., using the skipped layers (the 3,6,9,12-th) of the teacher for initialization, the performance is better than the baseline (62.9% vs. 64.4%), which further verifies the importance of representation similarity between the two ones.

**Determination of Eliminating Epochs.** We conduct experiments as shown in the Table 3g to choose the best number of epochs $m$ in the progressive eliminating period. We find that when the epoch $m$ greater than 40, the choice of $m$ seems hardly affect the performance. Accordingly we determine the epoch to be 40.

**Intermediate Teacher.** Intermediate teacher is introduced to promote the transferring capacity from a deep model to a shallow one. We conduct experiment as in Table 3f. We can observe that the intermediate teacher can bring a gain of 0.7% AUC score which can verify that.

### 4.2.4 Model Pruning Route

We present the model pruning route from the teacher model to MixFormerV2-B$^*$ and MixFormerV2-S in Tab. 3h and Tab. 3i respectively. The models on the first line are corresponding teacher models. We can see that, through the dense-to-sparse distillation, our token-based MixFormerV2-B obtains comparable accuracy with the dense-corner-based MixViT-B with higher running speed. Through the progressive model depth pruning based on the feature and logits distillation, MixFormerV2-B with 8 layers only decreases little accuracy compared to the 12-layers one.

### 4.3 Comparison with the Previous Methods

**Comparison with State-of-the-art Trackers.** We evaluate the performance of our proposed trackers on 6 benchmark datasets: including the large-scale LaSOT [20], LaSOT$_{ext}$ [20], TrackingNet [42], UAV123 [41], TNL2K [48] and VOT2022 [30]. LaSOT is a large-scale dataset with 1400 long videos in total and its test set contains 280 sequences. TrackingNet provides over 30K videos with more than 14 million dense bounding box annotations. UAV123 is a large dataset containing 123 aerial videos which is captured from low-altitude UAVs. VOT2022 benchmark has 60 sequences, which measures the Expected Average Overlap (EAO), Accuracy (A) and Robustness (R) metrics. Among them, LaSOT$_{ext}$ and TNL2K are two relatively recent benchmarks. LaSOT$_{ext}$ is a released extension of LaSOT, which consists of 150 extra videos from 15 object classes. TNL2K consists of 2000 sequences, with natural language description for each. We evaluate our MixFormerV2 on the test set with 700 videos. The results are presented in Tab. 4 and Tab. 5. More results on other datasets will be present in supplementary materials. Only the trackers of similar complexity are

| Method | LaSOT | | | LaSOT$_{ext}$ | | TNL2K | | TrackingNet | | | UAV123 | | Speed |
|---|---|---|---|---|---|---|---|---|---|---|---|---|---|
| | AUC | P$_{Norm}$ | P | AUC | P | AUC | P | AUC | P$_{Norm}$ | P | AUC | P | GPU |
| **MixFormerV2-B** | **70.6** | **80.8** | **76.2** | **50.6** | **56.9** | **57.4** | **58.4** | **83.4** | **88.1** | 81.6 | 69.9 | **92.1** | **165** |
| **MixFormerV2-B*** | 69.5 | 79.1 | 75.0 | - | - | 56.6 | 57.1 | 82.9 | 87.6 | 81.0 | **70.5** | 91.9 | **165** |
| MixFormer [14] | 69.2 | 78.7 | 74.7 | - | - | - | - | 83.1 | **88.1** | 81.6 | 70.4 | 91.8 | 25 |
| CTTrack-B [45] | 67.8 | 77.8 | 74.0 | - | - | - | - | 82.5 | 87.1 | 80.3 | 68.8 | 89.5 | 40 |
| OSTrack-256 [55] | 69.1 | 78.7 | 75.2 | 47.4 | 53.3 | 54.3 | - | 83.1 | 87.8 | **82.0** | 68.3 | - | 105 |
| SimTrack-B [7] | 69.3 | 78.5 | - | - | - | 54.8 | 53.8 | 82.3 | 86.5 | - | 69.8 | 89.6 | 40 |
| CSWinTT [46] | 66.2 | 75.2 | 70.9 | - | - | - | - | 81.9 | 86.7 | 79.5 | 70.5 | 90.3 | 12 |
| SBT-Base [49] | 65.9 | - | 70.0 | - | - | - | - | - | - | - | - | - | 37 |
| SwinTrack-T [34] | 67.2 | - | 70.8 | 47.6 | 53.9 | 53.0 | 53.2 | 81.1 | - | 78.4 | - | - | 98 |
| ToMP101 [39] | 68.5 | 79.2 | 68.5 | - | - | - | - | 81.5 | 86.4 | 78.9 | 66.9 | - | 20 |
| STARK-ST50 [52] | 66.4 | - | - | - | - | - | - | 81.3 | 86.1 | - | - | - | 42 |
| KeepTrack [40] | 67.1 | 77.2 | 70.2 | 48.2 | - | - | - | - | - | - | 69.7 | - | 19 |
| TransT [10] | 64.9 | 73.8 | 69.0 | - | - | 50.7 | 51.7 | 81.4 | 86.7 | 80.3 | 69.1 | - | 50 |
| PrDiMP [17] | 59.8 | 68.8 | 60.8 | - | - | - | - | 75.8 | 81.6 | 70.4 | 68.0 | - | 47 |
| ATOM [16] | 51.5 | 57.6 | 50.5 | - | - | - | - | 70.3 | 77.1 | 64.8 | 64.3 | - | 83 |

Table 5: State-of-the-art comparison on TrackingNet [42], LaSOT [20], LaSOT$_{ext}$ [20], UAV123 [41] and TNL2K [48]. The best two results are shown in **bold** and underline fonts. '*' denotes tracker with MixViT-B as the teacher during the dense-to-sparse distillation process. The default teacher is MixViT-L. Only trackers of similar complexity are included.

| Method | LaSOT | | | LaSOT$_{ext}$ | | TNL2K | | TrackingNet | | | UAV123 | | Speed | |
|---|---|---|---|---|---|---|---|---|---|---|---|---|---|---|
| | AUC | P$_{Norm}$ | P | AUC | P | AUC | P | AUC | P$_{Norm}$ | P | AUC | P | GPU | CPU |
| **MixFormerV2-S** | **60.6** | **69.9** | 60.4 | **43.6** | **46.2** | **48.3** | **43** | 75.8 | 81.1 | 70.4 | **65.8** | **86.8** | **325** | 30 |
| FEAR-L [5] | 57.9 | 68.6 | **60.9** | - | - | - | - | - | - | - | - | - | - | - |
| FEAR-XS [5] | 53.5 | 64.1 | 54.5 | - | - | - | - | - | - | - | - | - | 80 | 26 |
| HCAT[9] | 59.0 | 68.3 | 60.5 | - | - | - | - | **76.6** | **82.6** | **72.9** | 63.6 | - | 195 | **45** |
| E.T.Track [4] | 59.1 | - | - | - | - | - | - | 74.5 | 80.3 | 70.6 | 62.3 | - | 150 | 42 |
| LightTrack-LargeA [53] | 55.5 | - | 56.1 | - | - | - | - | 73.6 | 78.8 | 70.0 | - | - | - | - |
| LightTrack-Mobile [53] | 53.8 | - | 53.7 | - | - | - | - | 72.5 | 77.9 | 69.5 | - | - | 120 | 36 |
| STARK-Lightning | 58.6 | 69.0 | 57.9 | - | - | - | - | - | - | - | - | - | 200 | 42 |
| DiMP [3] | 56.9 | 65.0 | 56.7 | - | - | - | - | 74.0 | 80.1 | 68.7 | 65.4 | - | 77 | 15 |
| SiamFC++ [50] | 54.4 | 62.3 | 54.7 | - | - | - | - | 75.4 | 80.0 | 70.5 | - | - | 90 | 20 |

Table 6: Comparison with CPU-realtime trackers on TrackingNet [42], LaSOT [20], LaSOT$_{ext}$ [20], UAV123 [41] and TNL2k [48]. The best results are shown in **bold** fonts.

included, i.e., the trackers with large-scale backbone or large input resolution are excluded. Our **MixFormerV2-B** achieves state-of-the-art performance among these trackers with a very fast speed, especially compared to transformer-based one-stream tracker. For example, MixFormerV2-B without post-processing strategies surpasses OSTrack by 1.5% AUC on LaSOT and 2.4% AUC on TNL2k, running with *quite faster* speed (165 FPS vs. 105 FPS). Even the MixFormerV2-B with MixViT-B as the teacher model obtains better performance than existing SOTA trackers, such as MixFormer, OSTrack, ToMP101 and SimTrack, with much faster running speed on GPU.

**Comparison with Efficient Trackers.** For real-time running requirements on limited computing resources such as CPU, we explore a lightweight model, i.e. **MixFormerV2-S**, which still reaches strong performance. And it is worth noting that this is the first time that transformer-based one-stream tracker is able to run on CPU device with a real-time speed. As demonstrated in Figure 6, MixFormerV2-S surpasses all other architectures of CPU-real-time trackers by a large margin. We take a comparison with other prevailing efficient trackers on multiple datasets, including LaSOT, TrackingNet, UAV123 and TNL2k, in Tab 6. We can see that our MixFormerV2-S outperforms FEAR-L by a an AUC score of 2.7% and STARK-Lightning by an AUC score of 2.0% on LaSOT.

## 5    Conclusion

In this paper, we have proposed a fully transformer tracking framework MixFormerV2, composed of standard ViT backbones on the mixed token sequence and simple MLP heads for box regression and quality score estimation. Our MixFormerV2 streamlines the tracking pipeline by removing the dense convolutional head and the complex score prediction modules. We also present a distillation based model reduction paradigm for MixFormerV2 to further improve its efficiency. Our MixFormerV2 obtains a good trade-off between tracking accuracy and speed on both GPU and CPU platforms. We hope our MixFormerV2 can facilitate the development of efficient transformer trackers in the future.

# Acknowledgement

This work is supported by National Key R&D Program of China (No. 2022ZD0160900), National Natural Science Foundation of China (No. 62076119, No. 61921006), Fundamental Research Funds for the Central Universities (No. 020214380099), and Collaborative Innovation Center of Novel Software Technology and Industrialization.

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

## Appendix

### Broader Impact

In this paper, we introduce MixFormerV2, a fully transformer tracking approach for efficiently and effectively estimating the state of an arbitrary target in a video. Generic object tracking is one of the fundamental computer vision problems with numerous applications. For example, object tracking (and hence MixFormerV2) could be applied to human-machine interaction, visual surveillance and unmanned vehicles. Our research could be used to improve the tracking performance while maintaining a high running speed. Of particular concern is the use of the tracker by those wishing to position and surveil others illegally. Besides, if the tracker is used in unmanned vehicles, it may be a challenge when facing the complex real-world scenarios. To mitigate the risks associated with using MixFormerV2, we encourage researchers to understand the impacts of using the trackers in particular real-world scenarios.

### Limitations

The main limitation lies in the training overhead of MixFormerV2-S, which performs *multiple* model pruning based on the dense-to-sparse distillation and deep-to-shallow distillation. In detail, we first perform distillation from MixViT with 12 layers and plain corner head to MixFormerV2 of 12 layers. The 12-layers MixFormerV2 is pruned to 8-layers and then to 4-layers MixFormerV2 based on the deep-to-shallow distillation. Finally, the MLP-ratio-4.0 4-layers MixFormerV2 is pruned to the MLP-ratio-4.0 4-layers MixFormerV2-S for real-time tracking on CPU. For each step, it requires training for 500 epochs which is time-consuming.

### S.1 Details of Training Time

The models are trained on 8 Nvidia RTX8000 GPUs. The dense-to-sparse stage takes about 43 hours. The deep-to-shallow stage1 (12-to-8 layers) takes about 42 hours, and stage2 (8-to-4 layers) takes about 35 hours.

### S.2 More Results on VOT2020 and GOT10k

**VOT2020.**   We evaluate our tracker on VOT2020 [29] benchmark, which consists of 60 videos with several challenges including fast motion, occlusion, etc. The results is reported in Table 7, with metrics Expected Average Overlap(EAO) considering both Accuracy(A) and Robustness. Our MixFormerV2-B obtains an EAO score of 0.322 surpassing CSWinTT by 1.8%. Besides, our MixFormerV2-S achieves an EAO score of 0.258, which is higher than the efficient tracker LightTrack, with a real-time running speed on CPU.

**GOT10k.**   GOT10k [28] is a large-scale dataset with over 10000 video segments and has 180 segments for the test set. Apart from generic classes of moving objects and motion patterns, the object classes in the train and test set are zero-overlapped. We evaluate MixFormerV2 trained with the four datasets of LaSOT, TrackingNet, COCO and GOT10k-train on GOT10k-test. We compare it with MixFormer and TransT with the same training datasets for fair comparison. MixFormerV2-B improves MixFormer and TransT by 0.7% and 1.6% on AO respectively with a high running speed of 165 FPS.

| | KCF [26] | SiamFC [2] | ATOM [16] | LightTrack [53] | DiMP [3] | STARK [52] | TransT [10] | CSWinTT [46] | MixFormer [14] | **Ours-S** | **Ours-B** |
|---|---|---|---|---|---|---|---|---|---|---|---|
| **VOT20$_{EAO}$** | 0.154 | 0.179 | 0.271 | 0.242 | 0.274 | 0.280 | - | 0.304 | - | 0.258 | **0.322** |
| **GOT10k$_{AO}$** | 0.203 | 0.348 | 0.556 | - | 0.611 | 0.688 | 0.723* | 0.694 | 0.726* | 0.621* | **0.739*** |

Table 7: State-of-the-art comparison on VOT2020 [29] and GOT10k [28]. ∗ denotes training with four datasets including LaSOT [20], TrackingNet [42], GOT10k [28] and COCO [35]. The best results are shown in **bold** font.

### S.3 More Ablation Studies

**Design of Prediction Tokens.**   We practice three different designs of prediction tokens for the target localization in Tab. 8a. All the three methods use the formulation of estimating the probability

| token num. | MLP num. | AUC |
|:---:|:---:|:---:|
| 1 | 4 | 67.1% |
| 4 | 4 | 67.3 |
| 4 | 1 | 67.5% |

(a) **Different prediction designs**. 'token num.' indicates the number of the learnable prediction tokens, 'MLP num.' denotes the number of employed MLP layers for localization.

| Init. method | LaSOT | LaSOT_ext | UAV123 |
|:---:|:---:|:---:|:---:|
| Tea-fir4 | 62.9% | 45.2% | 65.7% |
| Tea-skip4 | 64.4% | 46.1% | 66.6% |
| PMDP | 64.8% | 47.1% | 67.5% |

(b) **Progressive Model Depth Pruning (PMDP).** 'Tea-fir4' denotes using first 4 layers of the teacher for student initialization. 'Tea-skip4' is using 4 skipped layers of the teacher.

Table 8: **More ablation studies**. The default choice for our model is colored in  gray .

distribution of the four coordinates of the bounding box. The model on the first line denotes using one prediction token and then predicting coordinates distribution with four independent MLP heads. It can be observed that adopting separate prediction tokens for the four coordinates and a same MLP head retains the best accuracy.

**More Exploration of PMDP**  Tea-skip4 is a special initialization method, which chooses the skiped four layers (layer-3/6/9/12) of the teacher (MixViT-B) for initialization. In other words, Tea-skip4 is an extreme case of ours PMDP when the eliminating epoch $m$ equal to 0. So it is reasonable that Tea-skip4 performs better than the baseline Tea-fir4, which employs the first four layers of the teacher (MixViT-B) to initialize the student backbone. In Table 8b, we further evaluate the performance on more benchmarks. It can be seen that ours PMDP surpasses Tea-skip4 by 1.0% on LaSOT_ext, which demonstrate its effectiveness.

**Computation Loads of Different Localization Head**  We showcase the FLOPs of different heads as follows. Formally, we denote $C_{in}$ as input feature dimension, $C_{out}$ as output feature dimension, $H_{in}, W_{in}$ as input feature map shape of convolution layer, $H_{out}, W_{out}$ as output feature map shape, and $K$ as the convolution kernel size. The computational complexity of one linear layer is $O(C_{in}C_{out})$, and that of one convolutional layer is $O(C_{in}C_{out}H_{out}W_{out}K^2)$.

In our situation, for T4, the Localization Head contains four MLP to predict four coordinates. Each MLP contains two linear layer, whose input and output dimensions are all 768. The loads can be calculated as:

$$Load_{T4} = 4 \times (768 \times 768 + 768 \times 72) = 2580480 \sim 2.5M$$

For Py-Corner, totally 24 convolution layers are used. The loads can be calculated as:

$$\begin{aligned}
Load_{Py-Corner} = 2 * (& 768 * 384 * 18 * 18 * 3 * 3 + \\
& 384 * 192 * 18 * 18 * 3 * 3 + \\
& 384 * 192 * 18 * 18 * 3 * 3 + \\
& 192 * 96 * 36 * 36 * 3 * 3 + \\
& 384 * 96 * 18 * 18 * 3 * 3 + \\
& 96 * 48 * 72 * 72 * 3 * 3 + \\
& 48 * 1 * 72 * 72 * 3 * 3 + \\
& 192 * 96 * 18 * 18 * 3 * 3 + \\
& 96 * 48 * 18 * 18 * 3 * 3 + \\
& 48 * 1 * 18 * 18 * 3 * 3 + \\
& 96 * 48 * 36 * 36 * 3 * 3 + \\
& 48 * 1 * 36 * 36 * 3 * 3) \\
= 3902587776 & \sim 3.9B
\end{aligned}$$

For simplicity, we do not include some operations such as bias terms and Layer/Batch-Normalization, which does not affect the overall calculation load level. Besides, the Pyramid Corner Head utilize additional ten interpolation operations. Obviously the calculation load of Py-Corner is still hundreds of times of T4.

### S.4 Visualization Results

**Visualization of Attention Map**    To explore how the introduced learnable prediction tokens work within the P-MAM, we visualize the attention maps of prediction-token-to-search and prediction-token-to-template in Fig. 5 and Fig. 6, where the prediction tokens are served as *query* and the others as *key/val* of the attention operation. From the visualization results, we can arrive that the four prediction tokens are sensitive to corresponding part of the targets and thus yielding a compact object bounding box. We suspect that the performance gap between the dense corner head based MixViT-B and our fully transformer MixFormerV2-B without distillation lies in the lack of holistic target modeling capability. Besides, the prediction tokens tend to extract partial target information in both the template and the search so as to relate the two ones.

**Visualization of Predicted Probability Distribution**    We show two good cases and bad cases in Figure 7. In Figure 7a MixFormerV2 deals with occlusion well and locate the bottom edge correctly. As show in Figure 7b, the probability distribution of box representation can effectively alleviate issue of ambiguous boundaries. There still exist problems like strong occlusion and similar objects which will lead distribution shift, as demonstrated in Figure 7c and 7d.

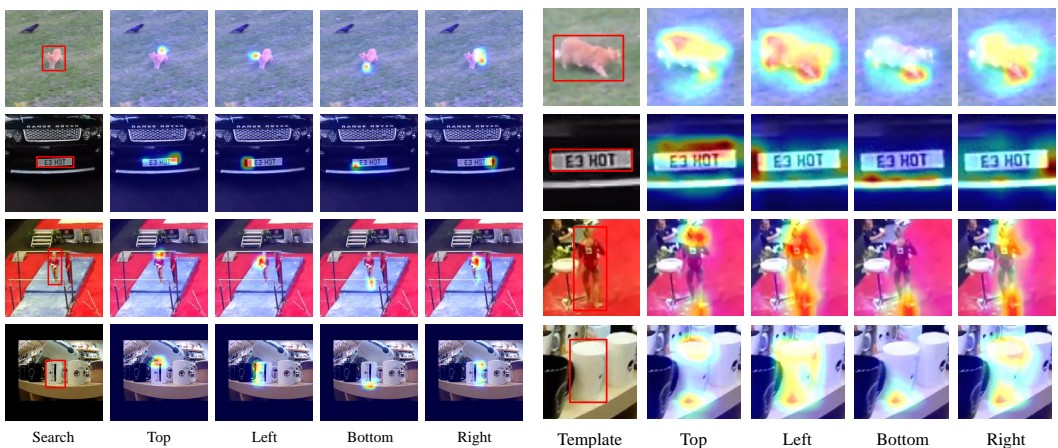

|  |  |  |  |  |
| Search | Top | Left | Bottom | Right |
| Template | Top | Left | Bottom | Right |

Figure 5: Visualization of prediction-token-to-search attention maps, where the prediction tokens are served as *query* of attention operation.

Figure 6: Visualization of prediction-token-to-template attention maps, where the prediction tokens are served as *query* of attention operation.

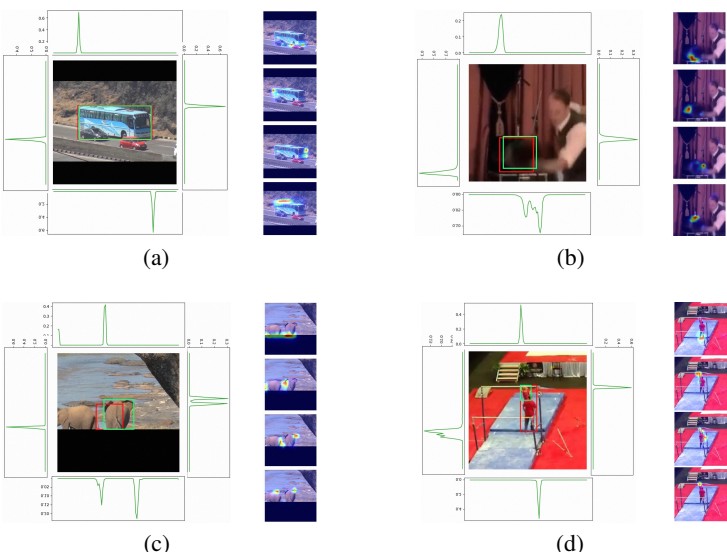

(a)    (b)

(c)    (d)

Figure 7: In each figure, the left one is plot of the probability distribution of predicted box (red), which demonstrates how our algorithm works. The right one is heatmap of attention weights in the backbone. The examples are from LaSOT test subset and the green boxes are ground truths.

