# OpenReview forum: "MixFormerV2: Efficient Fully Transformer Tracking"
_NeurIPS.cc/2023/Conference — NeurIPS 2023 poster_

### Official Review · Reviewer_eZL2 · 2023-07-03

**Soundness:** 3 good
**Presentation:** 3 good
**Contribution:** 2 fair
**Rating:** 5
**Confidence:** 5

**Summary:**

This paper proposes an efficient pure-Transformer-based tracking framework MixFormerV2. By replacing dense prediction heads and complex updating-score mapping modules with simple four-box tokens, the tracking pipeline is streamlined with high efficiency. Besides, Dense-to-Sparse Distillation and Deep-to-Shallow Distillation further alleviate the computation burden of transformer architecture. Experiments show that this method achieves a good trade-off between tracking accuracy and speed.

**Strengths:**

- The motivation that achieving efficient and effective tracking with Transformer is crucial and significant for the balance between performance and speed.
- Experimental results show that the proposed method effectively improves the inference speed while keeping excellent performance.
- The writing is clear and easy to follow.


**Weaknesses:**

- The proposed method is only applied on one baseline tracker MixFormer. The generality is not well proved on other Transformer trackers.
- More detailed analysis about the pruned parameters should be provided, e.g., the curve between the performance and prune ratio.


**Questions:**

- The training time of different distillation strategies is not provided, which reflects the practicability of the proposed method.
- Experiments evidence the effectiveness of teacher supervision with a similar architecture (i.e., MixFormer). I wonder if it works with the supervision of other Transformer trackers (e.g., OSTrack)?

---

> ### Author Rebuttal · Authors · 2023-08-10
>
> ***C1: The proposed method is only applied on one baseline tracker MixFormer. The generality is not well proved on other Transformer trackers.***
>
> **R1:** Thanks for your suggestion. To demonstrate the generality in rebuttal, we conduct experiments of **dense-to-sparse distillation** (i.e. the prediction token based head) and **deep-to-shallow distillation** on SimTrack[7]. Besides, we also validate the effect of deep-to-shallow distillation on OSTrack. we find that our proposed dense-to-sparse distillation can not be empolyed to OSTrack since the search tokens are gradually eliminated in some layers, which makes the four learnable prediction token hard to capture the object position information.
>
>
> |     Tracker     |   Head   |   Init   | LaSOT_ext | LaSOT |
> | :-------------: | :------: | :------: | :-------: | :---: |
> |  Sim-B/16(12)   | original |    -     |   48.8    | 68.6  |
> | Sim-B/16-T4(12) |  our T4  |    -     |   48.3    | 68.1  |
> | Sim-B/16-T4(8)  |  our T4  | our PMDP |   47.9    | 67.5  |
> | Sim-B/16-T4(8)  |  our T4  | Tea-fir8 |   46.2    | 66.0  |
>
> |     Tracker      |   Head   |   Init   | LaSOT_ext | LaSOT |
> | :--------------: | :------: | :------: | :-------: | :---: |
> | OSTrack-256 (12) | original |    -     |   47.4    | 68.8  |
> | OSTrack-256 (8)  | original | our PMDP |   46.9    | 67.9  |
> | OSTrack-256 (8)  | original | Tea-fir8 |   45.1    | 66.3  |
>
> Through the results in the above table (without carefully tuning hyper-parameters), we can derive that the proposed prediction token based architecture and the distillation method is effective on other one-stream transformer trackers.
>
> ***C2: More detailed analysis about the pruned parameters should be provided, e.g., the curve between the performance and prune ratio.***
>
> **R2:** Thanks for your useful advice, we provide the curve between the performance and prune ratio in the common response  pdf file. It can be observed that the proposed method can consistently improve the performance over baseline.
>
> ***C3: The training time of different distillation strategies is not provided.***
>
> **R3:**  Thanks for your suggestion. We will add the training time of different models in our revision. The models are trained on 8 Nvidia RTX 8000 GPUs. The dense-to-sparse stage takes about 43 hours. The deep-to-shallow stage1 (12 to 8) takes about 43 hours, stage2 (8 to 4) takes about 35 hours.
>
> ***C4: I wonder if it works with the supervision of other Transformer trackers (e.g., OSTrack)?***
>
> **R4:** We guess you means whether **dense-to-sparse** distillation could be supervised by other transformer trackers.
>
> For dense-to-sparse distillation, the proposed distribution-based regression is `ought to be supervised with corner prediction head, since it can provide the probability maps for top-left and bottom-right corners`, which has been illustrated in Section 3.2.1. OSTrack is not applicable as it does not have a coerner head . To respond to your concern, we use SimTrack of corner head as the teacher for this analysis. (Note that we did not carefully tune the hyper-parameters due to the limited time.)
>
> |  Tracker   | Teacher |  LaSOT |
> | :-------: | :-------:| :-------: |
> | MixFormerV2(12) | Sim-B/16 | 67.9 |
> | MixFormerV2(12) | MixViT-B | 68.9 |
>
> It concludes that SimTrack as a teacher can provide a comparable performance to MixViT-B.

---

> > ### Comment · Reviewer_eZL2 · 2023-08-20
> >
> > Some concerns have been addressed, so I keep my rating as borderline accept.

---

### Official Review · Reviewer_fR9F · 2023-07-05

**Soundness:** 3 good
**Presentation:** 3 good
**Contribution:** 3 good
**Rating:** 8
**Confidence:** 5

**Summary:**

Since main-stream tracking methods are somewhat limited in efficiency, this paper is motivated to develop efficient trackers. The proposed method (named MixFormerV2) is based on the recent tracker MixFormer. The key improvement of architecture is to use a token-based prediction head to replace the corner-based head. Four prediction tokens are used to predict the four coordinates of the bounding box, like the CLS token in ViT. In training, a dense-to-sparse distillation and a deep-to-shallow distillation are employed to further improve trackers’ efficiency. The dense-to-sparse distillation improve the performance of the token-based prediction head, while the deep-to-shallow distillation prune some layers of the backbone. Finally, the proposed MixFormerV2-B and MixformerV2-S achieve good trade-off of performance and speed on GPU and CPU.

**Strengths:**

1) The architecture of the model is simple. The main architecture is a ViT-based backbone with four prediction tokens.
2) The performance and speed is good. Compared with existing high-performance trackers, MixFormerV2-B achieves competitive performance with higher speed. Compared with existing high-speed trackers, MixformerV2-S achieves state-of-the-art performance.
3) Employing distillation is novel for tracking. The proposed dense-to-sparse distillation and deep-to-shallow distillation help the development of efficient tracking methods.

**Weaknesses:**

1) The description of Table 3(e) is not clear. Does Tea-skip4 use PMDP? If not, does its higher performance mean that PMDP is not necessary? If not, why can it achieve similar performance to PMDP?
2) The LaSOT extension benchmark is not included in comparison. The HCAT method is not included in the comparison.
3) The amount of parameters is not reported.

[1] Fan H, Bai H, Lin L, Yang F, Chu P, Deng G, Yu S, Huang M, Liu J, Xu Y, et al.. LaSOT: A High-quality Large-scale Single Object Tracking Benchmark. IJCV, 2021

[2] Chen X, Kang B, Wang D, Li D, Lu H. Efficient Visual Tracking via Hierarchical Cross-Attention Tracking via Hierarchical Cross-Attention Transformer. In ECCVW. 2022

**Questions:**

In rebuttal, I hope to see the response to question 1) in weaknesses. Besides, I suggest revising the paper according to 2) and 3) in weaknesses.

**Limitations:**

The paper does not provide a discussion on limitations. I recommend providing some failure cases, improvement aspects, etc.

---

> ### Author Rebuttal · Authors · 2023-08-10
>
> ***C1: Does Tea-skip4 use PMDP? If not, does its higher performance mean that PMDP is not necessary? If not, why can it achieve similar performance to PMDP?***
>
> **R1:** *Tea-skip4* is a **special initialization** method, which chooses the skiped four layers (layer-3/6/9/12) of the teacher (MixViT-B) for initialization. In other words, *Tea-skip4* is `an extreme case of ours PMDP when the eliminating epoch m equal to 0`. So it is reasonable that *Tea-skip4* performs better than the baseline *Tea-fir4*, which employs the first four layers of the teacher (MixViT-B) to initialize the student backbone, in the following table.
>
> We further evaluate the performance on the more challenging LaSOT_ext dataset. It can be seen that ours PMDP surpasses *Tea-skip4* by 1.0%, which demonstrate its effectiveness.
> |  Method   | LaSOT_ext | UAV123 | LaSOT  |
> | :-------: | :-------: | :----: | :----: |
> | Tea-fir4  |    45.2   |  65.7  |  63.0  |
> | Tea-skip4 |    46.1   |  66.6  |  64.4  |
> |   PMDP    |   **47.1**   |  **67.5**  |  **64.8**  |
>
> ***C2: The LaSOT extension benchmark is not included in comparison. The HCAT
> method is not included in the comparison.***
>
> **R2:** Thanks for your advice, we will add the following LaSOT_ext results and the HCAT comparison in our revision. As in the table, MixFormerV2-B surpasses SwinTrack-B and OSTrack-256 by a large margin with a high running speed.
>
> |  Method   | AUC | Norm P | P  |
> | :-------: | :-------: | :----: | :----: |
> | MixFormerV2-B  |   **50.6**  |  **61.0**  |  **56.9**  |
> | MixFormerV2-S  |  43.6   |  52.7  |  46.2  |
> | SwinTrack-B |    47.6   |  58.2  |  54.1  |
> | OSTrack-256  |   47.4  |  57.3  |  53.3  |
>
> ***C3: The amount of parameters is not reported.***
>
> **R3:** We will add the amount of parameters in the revised version.
>
> ***C4: The paper does not provide a discussion on limitations. I recommend providing some failure cases, improvement aspects, etc.***
>
> **R4:** We have provided the limitation in the supplementary material. In our revision, we will provide the visualization results of the failure cases and some improvement aspects. In the common response pdf file, we have also provided some visualization examples of the good cases and failure cases.

---

> > ### Comment · Reviewer_fR9F · 2023-08-20
> > **Re:**
> >
> > The authors have carefully addressed my concerns, thus, I update my rating as Strong Accept.

---

### Official Review · Reviewer_GURa · 2023-07-06

**Soundness:** 3 good
**Presentation:** 3 good
**Contribution:** 2 fair
**Rating:** 5
**Confidence:** 4

**Summary:**

This paper focuses on designing efficient transformer-based single object tracking algorithm. The basic idea is to replace the original corner-based convolution head with a more lightweight MLP head input w/ only 4 learnable tokens.  Moreover, both dense-to-sparse and deep-to-shallow distillations are designed for further improving the efficiency. Experimental results show that the proposed approaches achieve better speed-accuracy trade-off compared to existing transformer-based tracking approaches.

**Strengths:**

- The paper is well written and organized, which is easy to understand and follow.
- Using the distillation technique to achieve better speed-accuracy trade-off is technically sound.
- The proposed approaches are verified on various large-scale tracking benchmarks.
- The paper is solid in terms of both illustration and experimental results, which brings something new about how to perform distillation for speeding up existing transformer-based trackers.

**Weaknesses:**

- What’s the effect of  w/o applying the progressive strategy for depth pruning?
- It seems that the fast running speed mainly comes from the decrease of layers used in the transformer backbone as illustrated in Table 2. What’s the tracking performance and speed can be achieved if we only apply progressive model depth pruning and a corner prediction head (w/o using dense-to-sparse distillation)?
- Using the distillation technique to speed up deep models is not a very novel idea, since many attempts are working on distillation for speed up. One good thing is that the authors made some specifical design in VOT like a progressive pruning strategy in PMDP.
- The speed evaluation is somewhat questionable. In Tables 5-6, is the FPS comparison fair? is the speed (FPS) evaluated on the same platform? What is the GPU platform used for GPU speed and GFLOPs evaluation? Please detail these information in the main paper.

**Questions:**

No. See Weaknesses.

**Limitations:**

See Weaknesses.

---

> ### Author Rebuttal · Authors · 2023-08-10
>
> ***C1: What’s the effect of w/o applying the progressive strategy for depth pruning?***
>
> **R1:** As shown in the table, the baseline is the **most usual initialization** method, which employs the first four layers of the teacher (MixViT-B) to initialize the student backbone, denoted as *Tea-fir4*. On the challenging LaSOT_ext dataset, ours PMDP strategy improves it by 1.9% of AUC.
>
> Besides, we also explore an **special initialization** method, named *Tea-skip4*, which chooses the skiped four layers (layer-3/6/9/12) of the teacher (MixViT-B) for initialization. In fact, *Tea-skip4* is an extreme case of ours PMDP when the eliminating epoch $m$ equal to 0. It can be seen that ours PMDP also surpasses *Tea-skip4* by 1.0%, which demonstrate its effectiveness.
>
> |  Method   | LaSOT_ext | UAV123 | LaSOT  |
> | :-------: | :-------: | :----: | :----: |
> | Tea-fir4  |    45.2   |  65.7  |  63.0  |
> | Tea-skip4 |    46.1   |  66.6  |  64.4  |
> |   PMDP    |   **47.1**   |  **67.5**  |  **64.8**  |
>
> ***C2: It seems that the fast running speed mainly comes from the decrease of layers used in the transformer backbone as illustrated in Table 2. What’s the tracking performance and speed can be achieved if we only apply progressive model depth pruning and a corner prediction head (w/o using dense-to-sparse distillation)?***
>
> **R2:** As illustrated in Table 1, the proposed token-based head also improves the baseline by 84% of GPU running speed when using score prediction head for temporal-spatial tracking.
>
> The tracking performance and speed of using PMDP and a plain corner head (not pyramidal corner head) is showcased in the following table. It can be observed that MixFormerV2-B obtains comparable performance with higher running speed (+45 FPS).
>
> |  Backbone |  Head    | LaSOT | UAV123 | FPS(GPU)  |
> | :---------: | :---: | :---: | :----: | :-------: |
> | MixViT-8  |    Plain Corner   |  69.4  |  70.0 | 120 |
> | MixFormerV2-B  |    T4   |  69.5  |  70.5 | 165 |
>
> Excepting for the difference between corner head and the MLP head of MixFormerV2, our T4 design is more flexible. It allows us to easily predict the target confidence score with a simple MLP head, which is quite efficient compared to the original score prediction decoder (i.e., SPM) in MixFormer and is more easily deployed on different platforms (since the Precise RoI Pooling in SPM is not supported on CPU).
>
> ***C3: Using the distillation technique to speed up deep models is not a very novel idea.***
>
> **R3:** As you pointed, we have made some specifical design in VOT like the progressive pruning strategy and demonstrated its effectiveness. We think the exploration for distillation in VOT can bring some insprition to the tracking community.
>
> ***C4: In Tables 5-6, is the speed (FPS) evaluated on the same platform? What is the GPU platform used for GPU speed and GFLOPs evaluation?***
>
> **R4:** Thanks for your suggestions, and we will revise that in the next version of the manuscript. In Tables 5-6, the speed is evaluated on different platforms since we directly use the reported FPS in their original papers. In the table, we present the FPS of some transformer-based trackers on the same platform.
>
> |  Method |  FPS(GPU)    |
> | :---------: | :---: |
> | STARK-50 | 45 |
> | OSTrack-256 | 95 |
> | MixFormerV2-B | 165 |
>
> We use a NVidia Quadro RTX 8000 GPU for evaluation.

---

> > ### Comment · Reviewer_GURa · 2023-08-19
> >
> > I appreciate that the authors address most of my concerns in the rebuttal. Overall, I think this is a solid paper worth accepting for VOT, and I hope that the authors could also release their codes for reproduction, since there are still some training/inference hyper-parameters not included in the main paper. I will keep my previous rating.

---

### Official Review · Reviewer_iyBU · 2023-07-06

**Soundness:** 3 good
**Presentation:** 3 good
**Contribution:** 3 good
**Rating:** 6
**Confidence:** 5

**Summary:**

This paper proposes a fully transformer tracking framework without any dense convolutional operation and complex modules. The key contribution is the design of different input tokens and the distillation-based model reduction paradigm. Mixed attentions are performed between prediction tokens and the image generated tokens to capture abundant correlation. The distillation methods, dense-to-sparse distillation and deep-to-shallow distillation, improve the efficiency of the proposed MixFormerV2. Evaluations carried out on major tracking benchmarks show promising results.

**Strengths:**

- The proposed fully transformer tracking framework is interesting and makes sense.
- The proposed method achieves favorable performance in both terms of accuracy and run speed.


**Weaknesses:**

Some important details are missing.
- The details of some components are missing, such as the details of the Score Head.
- In the first ablation experiment, it shows that the run speed of the Py-Corner version is slower than that of T4. The calculation loads of T4 (under the VIT framework) is not light. It is suggested to provide a detailed calculation comparison between T4 and Py-Corner.
- Based on equation (8) and (9), some layers in the student network are gradually eliminated and finally turn into identity transformation. How to decide which layer to be eliminated? How to determine the epoch m in the training process. Are these eliminated parameters removed or just not involved in the process flow of the network.
- It is suggested to provide intermediate results, such as heatmaps, to better show how the proposed algorithm works.


**Questions:**

Please address the issues in the weakness section.

**Limitations:**

The authors do not discuss the limitations.

---

> ### Author Rebuttal · Authors · 2023-08-09
>
> ***C1:  The details of some components are missing, such as the details of the Score Head.***
>
> **R1:** We will add the detailed structure of the heads in our revision. The Score Head is a simple MLP composed of `two linear layers with the hidden dimension of 768.`
> Specifically, firstly we average these four prediction tokens to gather the target information, and then feed the token into the MLP-based Score Head to directly predict the confidence score $s$ which is a real number. Formally, we can represent it as:
> $$ s = \mathrm{MLP}(\mathrm{mean}(\mathrm{token}_X)), X \in \{\mathcal{T}, \mathcal{L}, \mathcal{B}, \mathcal{R}\}. $$
>
> ***C2: The calculation loads of T4 (under the VIT framework) is not light. It is suggested to provide a detailed calculation comparison between T4 and Py-Corner.***
>
> **R2:** Thanks for your suggestion, we will add the detailed comparison between Py-Corner head and T4 head in terms of FLOPs in our revised manuscrip. Since `the four prediction tokens can be concatenated to the search tokens as a whole for mixed attention in our implementation, the computational latency can be omitted`, which means only the four MLP heads are ought to calculate loads (FLOPs). Based on that, we showcase the FLOPs as follows.
>
> Formally, we denote $C_{in}$ as input feature dimension, $C_{out}$ as output feature dimension, $H_{in}, W_{in}$ as input feature map shape of convolution layer, $H_{out}, W_{out}$ as output feature map shape, and $K$ as the convolution kernel size.
> The computational complexity of one linear layer is $O(C_{in}C_{out})$, and that of one convolutional layer is $O(C_{in}C_{out}H_{out}W_{out}K^2)$.
>
> In our situation, for **T4**, the Localization Head contains four MLP to predict four coordinates. Each MLP contains two linear layer, whose input and output dimensions are all 768. The loads can be calculated as:
> $$Load_{T4} = 4 \times (768 \times 768 + 768 \times 72)= 2580480 \sim 2.5 M$$
> For **Py-Corner**, totally 24 convolution layers are used. The loads can be calculated as:
> $$ \begin{aligned}
> Load_{Py-Corner} = 2 * (&768 * 384 * 18 * 18 * 3 * 3 + \\\\
>                             &384 * 192 * 18 * 18 * 3 * 3 + \\\\
>                             &384 * 192 * 18 * 18 * 3 * 3 + \\\\
>                             &192 * 96 * 36 * 36 * 3 * 3 + \\\\
>                             &384 * 96 * 18 * 18 * 3 * 3 + \\\\
>                             &96 * 48 * 72 * 72 * 3 * 3 + \\\\
>                             &48 * 1 * 72 * 72 * 3 * 3 + \\\\
>                             &192 * 96 * 18 * 18 * 3 * 3 + \\\\
>                             &96 * 48 * 18 * 18 * 3 * 3 + \\\\
>                             &48 * 1 * 18 * 18 * 3 * 3 + \\\\
>                             &96 * 48 * 36 * 36 * 3 * 3 + \\\\
>                             &48 * 1 * 36 * 36 * 3 * 3) \\\\
>                       = & 3902587776 \sim 3.9 B
> \end{aligned}$$
> For simplicity, we do not include some operations such as bias term and Layer/Batch-Normalization, which does not affect the overall calculation load level. Besides, the Pyramid Corner Head utilize additional ten interpolation operations. Obviously the calculation load of Py-Corner is still hundreds of times of T4. In fact, `the FLOPs is not equal to running latency due to the parallel computing`, so we directly provide comparison of their running time in the manuscript.
>
> Another advantage of using T4 is the flexibility to allow us to easily predict the target confidence score with a simple MLP head, which is quite efficient compared to the original score prediction decoder (i.e., SPM) in MixFormer and is more easily deployed on different platforms (since the Precise RoI Pooling in SPM is not supported on CPU).
>
> ***C3: How to decide which layer to be eliminated? How to determine the epoch m in the training process. Are these eliminated parameters removed or just not involved in the process flow of the network?***
>
> **R3:** To embody multi-level representations and also reduce the difficulty of feature mimicking during elimination, we choose to eliminate layers uniformly. We have also experimented with eliminating all layers in high level or all layers in low level, but it turned out to perform worse than the employed choice.
>
> As shown in the table, we find that when the epoch $m$ greater than 40, the choice of $m$ seems hardly affect the performance. So we determine the epoch $m$ to 40.
> |     Arch      | Online | Epoch $m$ | AUC  |
> | :-----------: | :----: | :-------: | :--: |
> | MixFormerV2-B |   no   |    30     | 68.3 |
> | MixFormerV2-B |   no   |    40     | 68.5 |
> | MixFormerV2-B |   no   |    50     | 68.5 |
>
> These eliminated parameters can be removed directly when the drop rate decreased to zero in both training and inference processes.
>
> ***C4: It is suggested to provide intermediate results, such as heatmaps, to better show how the proposed algorithm works.***
>
> **R4:** Thanks for your advice, we will add the intermediate visualization results in our revised manuscript. We have provided the visualization results in the common response pdf.
>
> MixFormerV2 predict the probability distribution of bounding box coordinates, so we plot the output probability distribution as show in Fig. 1.
> We also visualize the attention heatmaps of four prediction tokens in the last layer of the backbone.
> It can be observe that the four prediction tokens focus on the four boundaries of the target object, and the coordinates probability distribution highly consists with the corresponding attention heatmaps.
>
> As shown in Fig. 1(c)(d), there still exist some cases the model is not able to handle perfertly, such as occlusion and similar objects which may cause distribution shift.
>
> ***C5: The authors do not discuss the limitations.***
>
> **R5:** We have discussed the limitations in the supplementary material, please check it.

---

> > ### Comment · Reviewer_iyBU · 2023-08-18
> > **Rebuttal by Reviewer**
> >
> >  Thank the authors for the compliment and detailed response. Thus, I suggest accepting this paper.

---

> > > ### Author Response · Authors · 2023-08-18
> > >
> > > Thanks a lot for your recognition of our work.

---

### Author Rebuttal · Authors · 2023-08-10

We thank all reviewers' efforts in reviewing our paper and giving insightful comments and valuable suggestions. We have provided the visualization of intermediate results and curve between performance and pruned depth in the PDF file.

---

> ### Comment · Area_Chair_MyPy · 2023-08-19
> **Reviewers please read and comment on authors' rebuttal**
>
> Dear reviewers,
> could you please read, respond to authors' rebuttals and evantually update your score?
> Regards

---

### Decision · Program_Chairs · 2023-09-21

**Decision:**

Accept (poster)

**Comment:**

The paper describes an approach for visual object tracking based on transformers. The motivation behind the paper is clear, that is redesigning the network to achieve a better balance between performance and speed. The reviewers mostly asked clarifications about the different components of the method and a few experimental results. The authors provided all the necessary clarifications. All reviewers agree that the paper can be accepted.